Global gap-analysis of amphipod barcode library

Jażdżewska Anna Maria anna.jazdzewska@biol.uni.lodz.pl 1
Tandberg Anne Helene S. 2
Horton Tammy 3
Brix Saskia 4
1 Department of Invertebrate Zoology and Hydrobiology, Faculty of Biology and Environmental Protection, University of Lodz , Lodz , Poland
2 University Museum, Department of Natural History, University of Bergen , Bergen , Norway
3 National Oceanography Centre , Southampton , United Kingdom
4 Department for Marine Biodiversity Research (DZMB), Senckenberg am Meer , Hamburg , Germany
Baird Donald
Electronic publication date: 2021 Nov 4
Publication date: 2021
Volume: 9
Electronic Location ID: e12352
Received 2021 Jun 22; Accepted 2021 Sep 29
Copyright: ©2021 Jażdżewska et al.
Copyright year: 2021
Copyright holder: Jażdżewska et al.
License: This is an open access article distributed under the terms of the Creative Commons Attribution License, which permits unrestricted use, distribution, reproduction and adaptation in any medium and for any purpose provided that it is properly attributed. For attribution, the original author(s), title, publication source (PeerJ) and either DOI or URL of the article must be cited.
License URL: https://creativecommons.org/licenses/by/4.0/

Keywords: DNA barcoding, Crustacea, Marine realm, Freshwaters, Semi-terrestrial, Taxonomic identification

Funding: The University of Lodz internal funds B2011000000069 Anna Maria Jażdżewska was supported by the University of Lodz internal funds (B2011000000069). The funders had no role in study design, data collection and analysis, decision to publish, or preparation of the manuscript.

==============================
In the age of global climate change and biodiversity loss there is an urgent need to provide effective and robust tools for diversity monitoring. One of the promising techniques for species identification is the use of DNA barcoding, that in Metazoa utilizes the so called ‘gold-standard’ gene of cytochrome c oxidase (COI). However, the success of this method relies on the existence of trustworthy barcode libraries of the species. The Barcode of Life Data System (BOLD) aims to provide barcodes for all existing organisms, and is complemented by the Barcode Index Number (BIN) system serving as a tool for potential species recognition. Here we provide an analysis of all public COI sequences available in BOLD of the diverse and ubiquitous crustacean order Amphipoda, to identify the barcode library gaps and provide recommendations for future barcoding studies. Our gap analysis of 25,702 records has shown that although 3,835 BINs (indicating putative species) were recognised by BOLD, only 10% of known amphipod species are represented by barcodes. We have identified almost equal contribution of both records (sequences) and BINs associated with freshwater and with marine realms. Three quarters of records have a complete species-level identification provided, while BINs have just 50%. Large disproportions between identification levels of BINs coming from freshwaters and the marine environment were observed, with three quarters of the former possessing a species name, and less than 40% for the latter. Moreover, the majority of BINs are represented by a very low number of sequences rendering them unreliable according to the quality control system. The geographical coverage is poor with vast areas of Africa, South America and the open ocean acting as “white gaps”. Several, of the most species rich and highly abundant families of Amphipoda (e.g., Phoxocephalidae, Ampeliscidae, Caprellidae), have very poor representation in the BOLD barcode library. As a result of our study we recommend stronger effort in identification of already recognised BINs, prioritising the studies of families that are known to be important and abundant components of particular communities, and targeted sampling programs for taxa coming from geographical regions with the least knowledge.

Introduction

Nature in the age of Anthropocene is facing numerous global changes and challenges. One of the drastic results of human associated activities is the acceleration of species extinctions, with one million species estimated to be presently critically endangered (IPBES, 2019). What is more, although the rate of species discovery grows, large numbers of species remain undescribed and it is believed many will not be recognized before they go extinct (Mora et al., 2011; Brix et al., 2020). This raises the challenge of efficient environmental monitoring, which is crucial for biodiversity recognition and preservation. Monitoring based on the taxonomic identification of organisms in samples is time-consuming and requires knowledge of the studied group. In the time of the taxonomic impediment (Ebach, Valdecasas & Wheeler, 2011), species identification methods offering an alternative to morphology-based methods are of great interest. Utilization of DNA-barcoding (identifying sequences of individual specimens), metabarcoding (high-throughput identification of bulk samples) and the use of environmental DNA (e-DNA, identifying DNA of taxa directly from water or soil sample, without collection of specimens) have been presented as promising methods in monitoring and ecological studies (e.g., Hajibabaei et al., 2012; Cristescu, 2014; Aylagas et al., 2018; Leese et al., 2018; Bush et al., 2019; Feio et al., 2020). The use of metabarcoding in assessing the status of ecosystems has already received the new term “Biomonitoring 2.0” (Bush et al., 2019). Such approaches require the existence of well-established barcode fragment libraries, which allow accurate recognition of organisms in the environment (Cristescu, 2014; Cowart et al., 2015; Oliveira et al., 2016; Múrria et al., 2020). Recent studies indicate that although the use of barcoding in biomonitoring has great advantages over morphological identification, the current gaps in barcode libraries may hinder their use (Weigand et al., 2019; Duarte, Vieira & Costa, 2020; Feio et al., 2020; Hestetun et al., 2020; Leite et al., 2020; Múrria et al., 2020; Vieira et al., 2021).

There are two main repositories where DNA sequences are deposited: NCBI GenBank (http://www.ncbi.nlm.nih.gov/genbank/, Sayers et al., 2020) and Barcode of Life Data System (BOLD, http://www.boldsystems.org, Ratnasingham & Hebert, 2007). In contrast to GenBank, which assembles nucleotide data of all genes, the primary aim of BOLD is to store data used for species barcoding, which in the case of Metazoa is the cytochrome c oxidase (COI) gene. The development of the BOLD database included the Barcode Index Number (BIN) system implementation (Ratnasingham & Hebert, 2013) that intends to help in biodiversity assessments by providing species-level taxonomic registry. Based on a molecular species delimitation method, each Molecular Operational Taxonomic Unit (MOTU) recognized by BOLD receives a unique alphanumeric code (BIN). Ideally, each BIN is associated with an accurate taxonomic (preferably species) identification and links to the voucher stored in a recognised institution. However, in practice this is not working well, and at the time of system implementation as many as 46% of BINs lacked species names (Ratnasingham & Hebert, 2013). This issue has arisen for a variety of reasons, which we investigate in this study using a particular faunal group, the Amphipoda, as a model.

The Order Amphipoda are peracarid crustaceans belonging to the class Malacostraca. They are very diverse components of aquatic environments. According to the World Amphipoda Database (WAD, Horton et al., 2020, accessed on 17-07-2020) there are 10,235 accepted amphipod species, the majority of which (78%) inhabit the marine realm, around 20% are freshwater species and just 2% are terrestrial taxa (Horton et al., 2020; Väinölä et al., 2008). The discovery rate of new species has grown steadily since the first amphipod species description and has particularly accelerated in the last six decades (Horton et al., 2020) with mean number of over 100 taxa annually described since the 1960s (Coleman, 2015). If the trend from the last sixty years persists, we may expect to have ca. 8,000 new species described by 2100. More conservative estimates predict that 6,100 new species will be described by that date (Arfianti, Wilson & Costello, 2018). The use of molecular methods in the studies of Amphipoda has revealed very high species diversity (e.g., Knox et al., 2012; Verheye, Backeljau & d’Udekem d’Acoz, 2016; Tempestini, Rysgaard & Dufresne, 2018; Jażdżewska & Mamos, 2019) and revealed the existence of cryptic species complexes within widely distributed taxa (Witt, Threloff & Hebert, 2006; Mamos et al., 2014; Wysocka et al., 2014; Havermans, 2016). Amphipoda are not only a species-rich group, but they also often dominate the crustacean assemblages in which they occur (e.g., Corkum, 1989; Humphries, Davies & Mulcahy, 1996; Vinogradov, Volkov & Semenova, 1996; Jazdzewski et al., 2001; Väinölä et al., 2008; Frutos, Brandt & Sorbe, 2017; Brix et al., 2018; Havermans & Smetacek, 2018). They can be found in both the benthos and the pelagic realm, presenting a variety of states of mobility (from epibenthic clingers to fully mobile swimmers) and, as a result, possess a wide variety of feeding habits including herbivory, detritivory, necrophagy, omnivory, predation and ectoparasitism (Barnard & Karaman, 1991; Vinogradov, Volkov & Semenova, 1996; Dauby, Scailteur & De Broyer, 2001; Väinölä et al., 2008). Being diverse and abundant they are important prey items for other invertebrates and vertebrates, including fish, birds and mammals (e.g., Dalpadado et al., 2001; Dauby, Nyssen & De Broyer, 2003). Certain species of Amphipoda are used in laboratory ecotoxicological studies (Hyne & Everett, 1998; Bundschuh et al., 2013; Major et al., 2013). Some amphipod species are well-adapted to anthropogenic environments such as artificial structures used in coastal protection or are part of fouling communities, and have shown a high invasion potential worldwide (e.g., Bij de Vaate et al., 2002; Kelly et al., 2006; Cabezas et al., 2014; Rewicz et al., 2015; Beermann et al., 2020; Sedano et al., 2020).

The combined factors of high diversity and the important role played by amphipods in the aquatic ecosystem highlight the need for accurate species identifications which are required for biological monitoring programs. The use of DNA-barcoding may speed up the identification process, but it will only succeed if the barcode library is well-established and robust. Recent gap-analyses of the barcode libraries in aquatic European environments showed very large differences in the coverage between different taxonomic groups and geographic regions (Weigand et al., 2019; Feio et al., 2020; Hestetun et al., 2020; Leite et al., 2020; Vieira et al., 2021). These studies used species lists restricted to particular geographic regions or chosen taxonomic groups. Basic summaries concerning the extent of amphipod data in BOLD identified problems with lack of taxonomic identification or detailed geographic information as well as contamination with human or bacterial DNA and provided recommendations to improve the data (Radulovici & Coleman, 2017; Coleman & Radulovici, 2020). However, to date there are no detailed analyses that have been conducted on a single taxon on a global scale.

In this study we have conducted a gap-analysis of the barcode library of a single crustacean order, the Amphipoda, on a global basis. In producing an up-to-date picture of the current state of knowledge, we will provide researchers with a detailed understanding of the both the strengths and the potential limitations of the use of DNA barcodes for identifications. We also propose recommendations for future initiatives that involve molecular data and produce new barcodes to fill the gaps in our knowledge of this taxon.

Material and Methods

Data for the present study were retrieved from BOLD by searching the “Public Data Portal” using the keyword “Amphipoda”. A combined dataset of all records was downloaded as an .xml file on June 24th 2020.

All records of the barcoding fragment of the cytochrome c oxidase I (COI-5P in BOLD) were extracted (29,016 records). This extracted dataset was used for all further analyses conducted by using various filtering options in an Excel spreadsheet. 2,579 records, represented by sequences shorter than 500 bp or having more than 1% ambiguous nucleotides for which BINs were not ascribed, were removed from dataset. Continued analysis of the dataset revealed some duplicate records (1,468 records, 734 cases, File S1). These derived from data harvested by BOLD from GenBank and seemed to be associated with an update of the records in GenBank. In the dataset, these records had an identical sample ID that referred to a GenBank Accession Number but with an additional ‘.1’ appended (e.g., KP713892 and KP713892.1) and with an identical identification provided. The differences were often linked with more detailed geographical information in the case of one record from the pair. Only the more detailed entry was retained for continued analysis. One sequence of Niphargus novomestanus S. Karaman, 1952 (KR858496, BOLD:ADD1128) was removed from the dataset because it was deleted from GenBank by its submitter (“This record was removed at the submitter’s request because the source organism cannot be confirmed.” GenBank website). The resulting dataset contained 25,702 records (Fig. 1, File S2).

Figure 1 PRISMA 2020 work-flow diagram (Page et al., 2021).

Summary of the data download, identification and screening before analysis. All record removals were done by the leading author of the paper.

Each record in the dataset was then further refined by sorting into categories according to the level of taxonomic identification. The following categories were used: order, family, subfamily, genus and species. Where records were provided with a temporary species identification, i.e., they are recognised as separate morphospecies but are not determined to correspond to a known taxon—they were treated as a separate category. In the whole dataset ca. 2.5% of records (596 individuals, 145 BINs) had uncertain identification with “cf.” or “aff”. Because the majority of them (417 records, 101 BINs) were associated with five species of one genus (Gammarus) for simplification all such records were treated as final species identifications. However, it is understood that the use of open nomenclature, when applied to identifications, provides an indication of the level of uncertainty, and may be intended to indicate the presence of new species or species complexes.

The data in BOLD come from wide variety of projects, some of which involve detailed taxonomic study by specialists, others are focused on monitoring or other topics in which taxonomic specialists are not involved. For the purposes of our analyses it was assumed that the identification accuracy was equal throughout the whole dataset, regardless of its origin. In several cases identification of the specimens within a single BIN varied strongly, with some records remaining at order level while others were determined to the species level. BINs aim to represent a putative species, so in the above example, the most detailed taxonomic information was applied to all records within the single BIN. Sometimes multiple (most often two) species or genus names were associated with a single BIN (87 cases). Each of these cases was checked individually. Sometimes it was an obvious misidentification of a single individual within a large group - if this was noted the misidentified record was added as an additional element to the records identified to the lowest congruent level (e.g., if the genus name matched the BIN genus, the misidentified taxon was added as an additional record identified to the genus level, if the lowest congruent level was family it was added to the family records); and the taxon identification of the majority of records was applied as correct. When it was impossible to judge which name was correct, the name of the identifier was checked and identifications carried out by taxonomists specializing in Amphipoda were prioritised over those provided by a non-specialist study. Where this process did not give a satisfactory conclusion, the BIN was allocated an identification at a rank that was congruent for the different records. The list of taxa with incongruent identifications together with an explanation of the final decision is presented in File S3.

Based on the taxonomic identification of the records the associated BINs were divided into the following environmental categories:

a) marine

b) freshwater

c) terrestrial.

Taxa that inhabit both marine realm and brackish environments were allocated to the marine category. Taxa from freshwater also occurring in brackish waters were allocated to the freshwater category. All representatives of the family Talitridae were treated as terrestrial taxa. Where taxonomic information was not detailed enough to provide environmental information about the particular BIN, the geographic data (coordinates and/or locality description) of the associated records were used to ascribe a particular BIN to one of the above categories. In some cases, this necessitated checking the original publication. A small number of unallocated BINs (18) and associated records (44) were used only in the first general summary of amphipod barcodes, but they were removed from further analyses (File S4).

In order to verify the correct environmental allocation of BINs, all BINs with records possessing coordinates were plotted on a map using the software QGIS2.16.1 (QGIS Development Team, 2018). Cases where incongruence between the ascribed environment and the geographic position appeared were checked individually. For those records without detailed geographic information the country of origin was taken from either BOLD or the associated publication.

In order to verify the barcode coverage within the studied group a list of BINs associated with a species name was compared with the list of accepted amphipod species names available in the World Amphipoda Database (WAD, Horton et al., 2020), accessed on 17-07-2020). A barcode quality assessment of the species represented in BOLD, based on the grading system proposed by Oliveira et al. (2016) and slightly modified by Fontes et al. (2020) was applied. This system consists of five grades: A – consolidated concordance (>10 sequences of a single morphospecies grouped in a single BIN), B– basal concordance (same as grade A but between three and 10 sequences available in the library), C– multiple BINs (one morphospecies assigned to more than one BIN), D– insufficient data (single species is assigned to single BIN but it is represented by less than three sequences in the barcode library), E– discordant species assignment (more than one species assigned to a single BIN). Fontes et al. (2020) provide an R-based application (Barcode, Audit & Grad System – BAGS), and uses only those records possessing species names. Since our aim was to focus on all available barcode records (including sequences identified only to higher ranks), the assessment was carried out manually. Additionally, as a result of initial treatment of the dataset, misidentified species records or BINs with unclear species identification, were already removed, so category E (discordant species assignment; Oliveira et al., 2016; Fontes et al., 2020) was not recorded. For the purpose of the present study Lysianassoidea incertae sedis was treated as an additional family. The amphipod families were divided into four categories depending on the number of species in each: low species rich families (up to 10 species), moderately species rich families (from 11 to 30 species), species rich families (31–100 species), very species rich families (more than 100 species). This division allowed verification of pattern between the species richness of the family and its representation in BOLD.

Results

Of the 25,702 amphipod COI records, 46.5% (11,958 records) were freshwater, 43.5% (11,169 records) were from the marine realm, and 9.8% (2,531 records) were terrestrial taxa. Of the 3,835 recognized BINs in total, 45% (1,726 BINs) belonged to freshwater taxa, 50% (1,920 BINs) were marine, and 4.5% (171 BINs) were from terrestrial taxa. 44 records (0.2%) and their associated 18 BINs (0.5%) could not be ascribed to the above environmental categories and were not considered further (Figs. 2A, 2B).

More than half (57.5%) of the records available in BOLD possessed coordinates, and 20% had information about the country of origin. Geographic information about the remaining 22.5% was provided only in the original publication. Geographic information is more comprehensive for marine taxa, where 71% of records possessed coordinates (compared to 47% for freshwater, and 50% for terrestrial taxa). Molecular studies of freshwater Amphipoda are focused mainly in the Northern hemisphere (particularly European countries, Russia and United States) while in the Southern hemisphere, Australia, New Zealand and Argentina are well studied (Fig. 3A). There is a complete lack of records (amphipod sequences) from Brazil, equatorial America and vast areas of Africa. Similar patterns of data coverage were seen for marine amphipods, which have greater numbers of records along European, North American and East Asian coasts. In the Southern hemisphere, Australia, New Zealand and Antarctica had larger numbers of barcode records (Fig. 3B). However, vast areas of the deep sea and the Arctic Ocean remain undersampled. Terrestrial Amphipoda in Europe, North America, China, Australia and Chile were the best represented (Fig. 3C), but sampling gaps were seen in the continents of South America and Africa.

Figure 2 Environmental origin of the amphipod records (A) and BINs (B) in BOLD database.

Figure 3 Geographic distribution of amphipod records expressed by sequences present in BOLD (A– freshwater, B– marine, C– terrestrial).

Dots indicate records with exact coordinates, for records without latitude and longitude the country of origin was checked. Background color of the country indicates this number per country.

The majority of records (69.8%, 17,922 recs.) had a complete species-level identification. Of the remaining 30.2% of records, 5.6% (1,433 recs.) had received temporary names (open nomenclature), 11.3% (2,902 recs.) remained identified at the genus level, 0.2% (40 recs.) at subfamily, 5.0% (1,285 recs.) at family, and 8.1% (2,076 recs.) at the order level. Levels of identification varied according to the environment, with marine taxa having greater proportions of taxa identified only to higher taxonomic ranks (Fig. 4A). The majority of BINs (3,817) were associated with species names (55.7%, 2,126 BINs). These were followed by BINs identified to the order level (13.3%, 506 BINs), generic or family level (10.7%, 407 BINs each) and those with a temporary name (9.4%, 359). BINs with only a subfamily name constituted just 0.3% (12). Greater variations between environments were seen for the BINs, with 74% (1,284) of freshwater BINs having a species level identification, compared to only 39% (751) of marine BINs (Fig. 4B). More than 20% (444, 23%) of the BINs for marine taxa remained identified at the order level.

Figure 4 Proportion of records (A) and BINs (B) with different level of identification within freshwater, marine and terrestrial amphipod taxa.

Regardless of the environmental origin, the majority of BINs were represented by a single sequence (Fig. 5). BINs represented by five or fewer sequences constituted around two thirds (67%, 114 terrestrial BINs to three quarters, 78%, 1,488 marine BINs) of BINs recorded in a particular environment. Freshwater taxa had 41 BINs (2.4%) represented by more than 50 sequences, compared to 28 (1.5%) for marine taxa, and eight (4.7%) for terrestrial taxa. When only those BINs with complete species-level identifications are considered, the proportion of sequences representing a particular MOTU does not change, with freshwater taxa having 78% of BINs (1,016) represented by five or fewer sequences. Almost three quarters of marine BINs (71%, 525 BINs) had five or fewer sequences in BOLD, while this proportion was 61% (56 BINs) for terrestrial taxa. Freshwater taxa had 35 BINs (3%) represented by more than 50 sequences, compared to 27 (4%) for marine taxa, and 6 (7%) for terrestrial taxa. The best represented BIN in BOLD (801 sequences) belonged to the terrestrial species Orchestoidea tuberculata Nicolet, 1849 (BOLD:ACQ3380), followed by the marine species Gammarus oceanicus Segestråle, 1947 (BOLD:AAA1262, 553 sequences), and the freshwater species Diporeia hoyi (S.I. Smith, 1874) (BOLD:AAA1473, 512 sequences). A further 26 BINs were represented by more than 100 sequences, including 17 freshwater, seven marine and two terrestrial BINs (File S5).

Figure 5 Number of BINs represented by given number of sequences.

Upper set (A, B, C)– all BINs, lower set (D, E, F)– only BINs with complete species-level identification considered. A, D– freshwater, B, E– marine, C, F– terrestrial taxa.

Out of the 3,817 studied BINs, just over half (55.7%, 2,126) were associated with a species-level identification, representing 1,001 species. Freshwater BINs with species identification reached 1,284, associated with 453 species, while 751 marine BINs were determined to 496 species. Of the 91 terrestrial BINs, 52 species were identified. Generally, a single morphological species was associated with each BIN (68%, 680 cases, 288 in freshwater, 359 marine, 33 terrestrial). 17% of the identified species were associated with two different BINs (72 freshwater, 82 marine and 14 terrestrial) (Fig. 6). There were however 19 cases when one single morphological species was represented by more than 10 BINs (17 freshwater, one marine and one terrestrial) (File S6). The greatest number of BINs was recorded for the freshwater species Gammarus balcanicus Schäferna, 1923 represented by 143 BINs (45 BINs were identified as “cf.” or “aff.”) followed by another freshwater taxon Hyallella azteca (Saussure, 1858) (62 BINs) and Gammarus fossarum Koch, 1836 (51 BINs; 19 BINs identified as “cf.” or “aff.”). Among terrestrial taxa the highest molecular variation (12 BINs) was recorded for Morinoia japonica (Tattersall, 1922) (present in BOLD under former generic name Platorchestia), while Apohyale stebbingi Chevreux, 1888 (with 11 BINs recognized) was the most diverse among marine species.

Figure 6 Number of nominal species represented by given number of BINs.

Of the 239 accepted families of Amphipoda (238 families and Lysianassoidea incertae sedis), 105 (44%) were represented by at least one species in BOLD (Table 1). The largest number of families had up to 20% of species barcoded, while only ten families had more than half of the known species barcoded (File S7). Thirteen families lacking barcoded species had at least one barcoded taxon identified at the genus level, a further five families had a taxon identified at the family level.

Just under ten percent (999 spp., 9.7%) of the 10,330 accepted species of Amphipoda (Horton et al., 2020) had barcodes. Of the nominal species possessing barcodes almost 500 (496 spp.) are marine, 451 spp. are freshwater and 52 spp. are terrestrial taxa. The data coverage of the majority of species, no matter their environmental origin, is not sufficient for the barcodes to be trusted according to the quality control system (Table 2) (Oliveira et al., 2016; Fontes et al., 2020). Additionally, a large group of taxa is represented by multiple BINs; only 10% of species represent consolidated concordance of available barcodes.

Table 1 Representation of amphipod families in BOLD.

Number of families		
without any barcoded species	117 (+ 13g, 5f)*	
with up to 10% barcoded species	47	
with 11–20% barcoded species	24	
with 21–50% barcoded species	24	
with >50% barcoded species	10	
Notes.

* in parentheses the number of families without barcoded species but with at least one BIN identified to the genus (g) or family (f) level.

The breakdown of amphipod families according to the assigned categories of richness and their respective representation in BOLD can be seen in Table 3. Almost every one of the very species rich families had at least one species barcoded (31 families out of 32), and 22 of 30 species rich families are represented in BOLD. For both moderately low and low species rich families 26 possessed at least one representative in BOLD constituting respectively 48% and 21% of all families each (File S7). The mean coverage of barcodes for species in each of the above groups was around 10% with the highest observed for low species rich families (12%) and the lowest (8%) recorded for families grouping from 30 to 100 species. However, if the families without any molecular information were removed from the study these numbers considerably change. The low species rich families (1–10 spp.) had a barcode coverage at the level of 49%, moderately species rich families (11–30 spp.) reached 21% of coverage, while the rich and very rich amphipod families (more than 30 spp.) had only 9–10% of species studied.

Table 2 Number of amphipod species in each realm with indication of their barcode quality according to grading system from Fontes et al. (2020).

A–consolidated concordance, B– basal concordance, C– multiple BINs for single morphospecies, D– insufficient data; for more detailed explanation of grading system, see Material and Methods section.

	A	B	C	D	All species	
All species	100	155	276	468	999	
Freshwater spp.	31	55	140	225	451	
Marine spp.	58	92	120	226	496	
Terrestrial spp.	11	8	16	17	52	

Table 3 Number of accepted families and species of Amphipoda (according to WAD accessed on 17-07-2020), number of families with representation in BOLD, number of species present in BOLD and mean coverage of barcodes in amphipod families represented in BOLD.

	No. of families	No. of species	No. of families with species representation in BOLD	No. of species present in BOLD	Mean barcode coverage [%] of those families with representation in BOLD	
Very species rich families (>100 spp.)	33	7,302	32	714	8	
Species rich families (31–100 spp.)	30	1,633	22	127	10	
Moderately species rich families (11–30 spp.)	53	979	26	107	21	
Low species rich families (<10 spp.)	123	416	26	51	49	

A third of families (34) have at least one species characterized by consolidated concordance of available barcodes (category A of the quality grading system). Another third of families (38) do not have any species in categories A or B, indicating that the species already studied represent a potential cryptic diversity or the available data are insufficient (Table 4).

Table 4 Percent of families with species belonging to different quality grading categories (Fontes et al., 2020).

A– consolidated concordance, B– basal concordance, C– multiple BINs for single morphospecies, D– insufficient data; for more detailed explanation of grading system, see Material and methods section.

	% of families	
	All families	Very species rich families (>100 spp.)	Species rich families (31–100 spp.)	Moderately species rich families (11–30 spp.)	Low species rich families (<10 spp.)	
At least one sp. in the category A	32.4	65.6	31.8	16	7.7	
At least one sp. in the category B	31.4	28.1	36.4	28	34.6	
At least one sp. in the category C	10.5	0	13.6	24	7.7	
At least one sp. in the category D	25.7	6.3	18.2	32	50	
Number of families	105	32	22	25	26	

Within the very species rich families, the best representation in BOLD was recorded for Niphargidae (36.5% of known species represented with a barcode), Gammaridae (31%) and Crangonyctidae (16%). Only the family Stegocephalidae did not have any representative with species level identification (although barcodes belonging to this family but identified at genus level were present). The least studied families within this group (but having at least one species barcode) were: Phoxocephalidae (1% of the species with a barcode), Dexaminidae, Liljeborgiidae and Maeridae (ca. 2% of the species with a barcode). Among species rich families 41% of the species from Pseudoniphargidae had barcodes, while the Epimeriidae and Pontogammaridae had 20% and 19%, respectively. The best represented moderately species rich families were Metacrangonyctidae, Oxycephalidae and Hyperiidae with 55%, 50% and 48% of the associated species represented with a barcode respectively. Within low species rich families four (Baikalogammaridae, Crymostygidae, Cyllopodidae and Tryphanidae) had all known species represented with barcodes, but other than Cyllopodidae (two species) the families are monotypic (File S7).

Discussion

Extent of barcode library of Amphipoda

One of the aims of establishing the BOLD database was to store and publish barcodes, based on records uploaded by its users and supplemented by the data harvested from GenBank (Ratnasingham & Hebert, 2007). Together with the BIN system, that groups similar sequences in clusters representing putative species (Ratnasingham & Hebert, 2013), the BOLD database aids in recognising and quantifying biodiversity. The extent of data in BOLD expresses the activity of researchers studying particular groups using molecular methods. The number of available sequences of Amphipoda in BOLD is comparatively large. At the time of download (end of June 2020) they were represented by almost 26,000 records (3,835 BINs), and by the end of August there were more than 34,000 public sequences (3,914 BINs) (BOLD accessed on 20-08-2020), indicating the great intensity of molecular studies involving this crustacean group, and that the data in BOLD are actively growing. Among other crustacean groups only Decapoda is represented by a higher number of records (64,281 records). Copepoda are represented by 18,511, Thecostraca by 15,554, Isopoda by 13,858 and Branchiopoda by 12,326 sequences. The large number of identified BINs within the Amphipoda also places this group second only to Decapoda (with 6,056 BINs). Isopods and copepods are represented by 1,853 and 1,804 BINs, respectively, while 969 BINs were identified within Branchiopoda. Within Thecostraca only 545 BINs were identified (boldsystems.org, accessed on 20-08-2020).

When the BIN system was implemented, Ratnasingham & Hebert (2013) indicated that 12% of all the sequences available in BOLD lacked a family name, 19% a genus name and 40% a species name. A comparison of these numbers with the present data on Amphipoda looks optimistic, where only 8% of sequences are without family indication, 13% are without genus and 29% lack a species identification. However, the global analysis of Ratnasingham & Hebert (2013) identified 10% of BINs lacking family names, almost 24% lacking generic names and 46% lacking species names. These numbers are almost identical for amphipod BINs known presently (13%, 23%, 43% of BINs lacking family, genus and species information, respectively). Among all known species of Amphipoda, almost 80% of species are marine, some 20% live in freshwaters, while 2% may be considered as terrestrial (Horton et al., 2020; Väinölä et al., 2008). The above proportions are expressed neither in the number of records nor the number of recognized BINs that are more or less evenly distributed between freshwater and marine taxa. This demonstrates that in terms of amphipod crustaceans, freshwater taxa are much better studied than the marine taxa. These disproportions are even more striking when the level of identification of sequences and BINs is considered. Although the majority of data in BOLD possess species-level identifications, marine amphipods are less thoroughly identified. This is especially clear for marine BINs, of which only 39% had species-level identifications, while as much as one fifth are identified only as “Amphipoda”. The fact that freshwater amphipods are better studied is not surprising considering the easier access to this environment. In the case of marine fauna, obtaining samples suitable for molecular analysis can be challenging, especially when extreme habitats (polar regions, deep-sea, hydrothermal vents etc.) are considered (e.g., Riehl et al., 2014; Jażdżewska & Mamos, 2019). Additionally, rarity is a common feature of numerous marine species (particularly in the deep-sea environment, see Kaiser, Barnes & Brandt (2007)), where many taxa are known only from their original descriptions and type localities (Jażdżewska & Mamos, 2019). The question of how many of the BINs not associated with a species identification actually belong to already known species is also of concern. In these cases, it is highly advisable to put every effort to identify the already available material; this will relatively efficiently improve data usability. Taxa that are associated with a BIN, yet are known to be new to science are another cause for concern. This is particularly evident for marine taxa collected during recent deep-sea exploration programs (e.g., Brandt et al., 2007; Jażdżewska, 2015; Brandt et al., 2019; Brix et al., 2020). It is imperative that full scientific descriptions of new species are produced to reduce the current proliferation of ‘dark taxa’ (Page, 2016).

The geographic distribution of available amphipod sequence records shows clear sampling gaps. In particular the African continent, the northern part of South America and the Coral Triangle in Asia are complete “white spots” when freshwater and terrestrial taxa are considered. For marine species, the coasts of Africa and South America, the Coral Triangle, and large parts of the deep sea of all oceans, lack coverage. Considering the known high species diversity of these regions it will be necessary to establish targeting sampling programs before we can consider that we have adequate global coverage of the molecular diversity of the Amphipoda.

Our study shows that globally the barcoding coverage of amphipod species is only about 10%. In comparison, just over 20% of all species registered in the European Register of Marine Species (ERMS) and almost 50% of species listed in the AZTI Marine Biotic Index (AMBI) have been barcoded (Weigand et al., 2019). The percentage of barcoded European freshwater invertebrates used in environmental monitoring reaches 64.5%, and when considering only Peracarida, 24% of ERMS species, 45% of AMBI and 82% of freshwater monitored taxa have been barcoded (Weigand et al., 2019). It has to be emphasized however, that only ERMS lists all marine invertebrates from the European region, while both of the other datasets studied by Weigand et al. (2019) consist of a subset of species from this area. More specific studies of Iberian macroinvertebrates revealed that ca. 40% of amphipod species possess barcodes (Leite et al., 2020; Múrria et al., 2020). Hestetun et al. (2020) conducted a barcode library gap-analysis of the benthic macrofauna of one region of the North Sea, which indicated the barcode coverage varying from 42.4% to 61% (depending on the calculation method) while Vieira et al. (2021) found that in Macaronesia 34.2% to 72.6% of macroinvertebrate species have barcode representation with much better coverage of non-indigenous taxa in comparison to the native ones. This indicates that for smaller subset of taxa and specified geographic region it is much easier to produce good barcode coverage. It can be concluded that although Amphipoda are an actively studied taxonomic group where scientists increasingly use molecular methods, this diverse and abundant macrofaunal taxon is still insufficiently represented in the BOLD barcode library.

Quality of amphipod barcode library

In order to provide a trusted barcode for a particular species, at least one good quality sequence associated with a species-level identification provided by taxonomic specialist is required as an absolute minimum. However, a single sequence cannot provide information about intraspecific variation, and overlooked contamination of the sample will mean the sequence cannot be validated. As such, it is advisable to provide a small number of sequences to characterise each taxon. The recently proposed barcode quality auditing system suggests providing at least three sequences to enable proper barcode evaluation (Oliveira et al., 2016; Fontes et al., 2020). Unfortunately, as we have shown in the case of Amphipoda, globally more than half of BINs are represented by only 1–2 sequences in BOLD. This low number of sequences places them in category D of the Oliveira et al. (2016) system, indicating the existing data is insufficient for use as trusted barcodes. Similar observations for a restricted amphipod dataset are made by Fontes et al. (2020).

Due to methodological differences it is impossible to make direct comparisons of our data with the results of the gap analysis of aquatic organisms in European waters (Weigand et al., 2019), but re-calculation of their data shows much improved barcode coverage. Among all freshwater invertebrates 65% of taxa barcoded are represented by more than five sequences, while this percentage rises to 77% when considering only freshwater Peracarida. This proportion of high quality datasets diminishes when marine taxa are considered; with 52% of the marine species from the AMBI list and 45% those listed in ERMS having at least five barcodes available. These numbers do not change when considering only marine Peracarida (52% and 46% of the ones presented in AMBI and ERMS lists, respectively). Our analysis of Amphipoda shows the opposite pattern with about 1/4-1/3 of BINs represented by more than five sequences but the good barcode coverage observed by Weigand et al. (2019) may be biased by the fact that they targeted the species used in water quality assessment programs. Because of their practical use such taxa receive more scientific interest and it may be assumed that their barcoding is prioritized by different institutions.

The amphipod BINs that have the largest numbers of sequences in BOLD are often the result of detailed studies of targeted species, which have produced large numbers of sequences as a secondary aim of the study. For example, 750 out of the 801 sequences in BOLD of terrestrial Orchestoidea tuberculata come from a single study by Brante et al. (2019); 406 records out of 411 sequences in BOLD of freshwater Dikerogammarus haemobaphes (Eichwald, 1841) come from Jażdżewska et al. (2020); while 232 records of 235 sequences in BOLD of marine Caprella scaura Templeton, 1836 come from Cabezas et al. (2014). The disproportional representation between the few species that are very thoroughly studied, and the remaining majority of species that are represented only by a single, or a low number of sequences, emphasises the need for more targeted sampling of less common species.

Best studied families and cryptic diversity

Almost half of the 239 known amphipod families are represented in BOLD. However, only ten of these families have more than 50% of their associated species sequenced. It is important to underline that there are 18 families in BOLD that do not have species-level identifications, but have records left at the family or genus level. A small effort to provide trusted species-level identifications for these taxa will greatly improve barcode coverage of the Amphipoda, particularly if they represent species already known to science.

Another concern that has arisen as part of this study relates to the format of temporary names in GenBank and BOLD, the different requirements by users for their input, and how this has changed following development of the databases. In GenBank, the incorporation of temporary names or codes is allowed (referred to as placeholder names in GenBank). In 2010, a large amount of COI data was incorporated into the BOLD database. The identifications associated with each of these imported sequences were included verbatim from GenBank. BOLD users, however, were originally able to use temporary names in the database only in private projects/dataset and when opening their data for public they were expected to provide the identification to the lowest taxonomic level possible (e.g., genus) and to provide the temporary name (e.g., incorporating “cf.” or “aff.”) as a taxonomy note (that has happened to the authors of the present paper). However, in BOLD a taxonomy note is only visible when the specimen page is open, and not in a general search. Recently, we have learnt that open nomenclature identifiers (such as ‘cf.’ and ‘aff.’) are now accepted by BOLD, but it may be assumed that numerous records remain at a higher taxonomic level, with more detailed identifications available that are hidden from general searches. This discrepancy in dealing with temporary names has become apparent when analysing the whole dataset as part of this study. In particular, the inconsistent use of temporary names in these databases mean that it is very difficult to differentiate between temporary names which are being used to refer to species that are new to science, and those which have remained at a higher taxonomic level because they were simply not identified further (which could be for a variety of reasons). Molecularly well-defined temporary names for new species are likely to become more abundant and therefore critical to our knowledge of biodiversity in the coming years, and we need to ensure they are managed carefully and consistently. Recommendations for the use of open nomenclature have been proposed recently to attempt to standardise and overcome these issues (Sigovini, Keppel & Tagliapietra, 2016; Horton et al., 2021) and it is hoped that these standard formats will be considered for use in both BOLD and GenBank.

Barcode coverage of families varies depending on the species richness. For species rich families it is around 10%, while coverage is increased for moderate and low species rich taxa. This is not surprising considering it is much easier to receive better coverage for monotypic families or those represented by only a few species. The best studied families are the ones that remain under the interest of large working groups who focus on studying specific families (e.g., Hou, Fu & Li, 2007; Mamos et al., 2014; Wysocka et al., 2014; Delić et al., 2017a; Delić et al., 2017b; Fišer et al., 2017; Copilaş-Ciocianu, Sidorov & Gontcharov, 2019). It is worth noting that providing barcodes is generally more a “by-product” of other analyses than the goal per se. Another issue that should be emphasized is that species rich families are proportionally under studied. This is important because they usually do not only group many species but very often the species from these families constitute the majority of amphipods characterizing different assemblages. This is clearly shown by the Phoxocephalidae (1% of the 367 known species are barcoded), Ampeliscidae (7% of 312 of the known species barcoded) or Oedicerotidae (10% of the 246 known species barcoded), all constitute very large and important components of marine benthic communities worldwide (Brandt, 1993; Weisshappel & Svavarsson, 1998; Frutos & Sorbe, 2017; Brix et al., 2018). Another example is provided by the Caprellidae (6% of the 443 known species are barcoded) which are an important part of many fouling communities (e.g., Ros, Vázquez-Luis & Guerra-García, 2013; Ros et al., 2013) and where proper species identifications are crucial in the context of growing transport with their resulting potential alien species invasions (op. cit.). The studies of these families should be prioritized in order to support marine monitoring programs based on barcode libraries.

The analysis of the amphipod BINs with a species-level identification showed that there were only a few cases where multiple names were associated with a single molecular unit. A quarter of these cases resulted from the misidentification of single individuals within a taxon. In some cases different names were associated with the description of new species (present in the database under both former and newly established name). Problems with morphological identification of cryptic species and the lack of well-established diagnostic characters within closely related species may also be the reason for the presence of multiple names for single BIN. The above problems have been recognized within Gammarus ochridensis Schäferna, 1926 species complex that is the group of morphologically very similar species of which two Gammarus cryptosalemaai Grabowski, Wysocka & Mamos, 2017) and Gammarus cryptoparechiniformis Grabowski, Wysocka & Mamos, 2017 are recognizable only based on molecular data (Wysocka et al., 2013; Grabowski, Wysocka & Mamos, 2017). This indicates that generally BOLD may be considered a trusted tool for species identification. Our analyses showed that in the majority of cases, a single BIN was characterising a single species, which is congruent with the results of other similar studies (Fontes et al., 2020; Leite et al., 2020). Some morphologically identified species were represented by two or even three BINs, which can indicate overlooked diversity. It has been noted however, that sometimes due to the methodology used during BIN-identification and the threshold used (2% of similarity, Ratnasingham & Hebert, 2013) some valid species may be split into two or more BINs (Lörz, Jażdżewska & Brandt, 2018; Jażdżewska & Mamos, 2019). This happens more frequently when the sample size is small and the intraspecific variation range cannot be adequately assessed. In such cases, the use of additional genes or other data analysing methods may help to decide the proper species delineation. The present study revealed 19 morphological species that were represented by 11 or more BINs. This multi-BIN representation was much more common in freshwater environments, where 17 species with potential cryptic diversity were observed. The existence of such high cryptic diversity especially in European waters was recognized by authors of the original works (Witt, Threloff & Hebert, 2006; Bauzà-Ribot et al., 2011; Knox, Hogg & Pilditch, 2011; Major et al., 2013; Mamos et al., 2014; Wysocka et al., 2014; Delić et al., 2017a; Delić et al., 2017b; Fišer et al., 2017; Tomikawa et al., 2017) most recently confirmed also by Wattier et al. (2020). A detailed study of the available barcodes and cryptic diversity of the Gammaridae and other representatives of the superfamily Gammaroidea is in preparation (T. Mamos, 2021, pers. comm.). The large representation of freshwater taxa forming cryptic species complexes (especially in Europe) can be partly explained by the geological events that shaped the European freshwater system (Wysocka et al., 2014; Mamos et al., 2016; Wattier et al., 2020). Presence of marine cryptic or pseudo-cryptic species have also been reported (Havermans, 2016; Verheye, Backeljau & d’Udekem d’Acoz, 2016), but the extent of molecular studies of amphipods from this realm is much smaller and as a result cryptic species may have been overlooked. A study of the marine genus Apohyale showed high diversification of species within the genus, and confirms that more studies are required to correctly identify species diversity and uncover cryptic diversity in marine taxa (Desiderato et al., 2019). Cases of highly diverse nominative species usually come from studies based in a single research group that was already aware of the high diversity within the taxon. There are, however, cases where multiple BINs have received the same identification but this was carried out by different authors at different times (without comparison of the material) and it is difficult to judge if the observed diversity is a result of the existence of a cryptic species or of misidentification of the species. In such cases it is impossible to decide which of the BINs represents the known species and which are cryptic/new species that require more detailed study (Jażdżewska et al., 2018). A detailed analysis of the species represented by several BINs was not the focus of the present study, but it should be a priority for BOLD to identify such cases and inform users about the presence of possible cryptic taxa. Users of BOLD who seek to obtain identification for their own sequence should be notified that the specimen they have may belong to a group of cryptic species and that the taxonomic identification should be treated with caution. Some initiatives to improve the curation of BOLD data have already begun (Radulovici et al., 2021) and it is highly recommended that mistakes or problematic issues that are found in the database are corrected and published e.g., the case of Hyperiella antarctica Bovallius, 1887/H. dilatata Stebbing, 1888 which was recently clarified by Havermans et al. (2019).

Conclusions and future recommendations

We have conducted a gap-analysis of the barcode library using a single crustacean order, the Amphipoda, as a model. The high diversity and the important role played by amphipods in the aquatic ecosystem combine to highlight the need for accurate species identifications which are required for biological monitoring programs. DNA-barcoding may speed up the identification process, but success is dependent on the barcode library coverage and quality. Our gap analysis has shown that although a large number of BINs (indicating putative species) was recognized by BOLD still only 10% of the amphipod species are represented by barcodes. Moreover, most BINs are represented by a very low number of sequences making them unreliable according to the quality control system. The geographical coverage is poor with vast areas of Africa, South America and the open ocean acting as “white gaps”, also the level of barcoding effort is skewed depending on the environment.

As such, we make the following recommendations (in order of priority), which will improve the data currently held within BOLD, and we outline steps that are needed to provide a more equal coverage of the sequence data within the Amphipoda, and thus improve the utility of the database for a variety of applications, including species identification and biomonitoring.

1. Morphological identification of the already recognised BINs (that are missing species ID) if the voucher specimens are available.

2. Analysis of the nominal species that are represented by more than one BIN, especially if identifications represented by different BINs were produced by separate working teams.

3. Prioritised barcoding of representatives from families that are known to be important and abundant components of communities; Phoxocephalidae, Ampeliscidae, Oedicerotidae, and Caprellidae should be prioritised.

4. Targeted sampling programs for taxa coming from geographical regions with the least knowledge.

5. Targeted sampling to obtain more sequences for taxa present in BOLD but represented by small numbers of sequences (especially singletons), from different parts of the species’ range if possible.

6. Targeted programs to sequence type specimens stored in musea or to collect and study fresh individuals from type localities if types are unsuitable for analyses.

Supplemental Information

Supplemental Information 1 List of the doubled records indentified in the BOLD database

Click here for additional data file.

Supplemental Information 2 The list of records analysed (after removal of doubled records)

The colors of the cell of recordID indicates the environment: green - marine (including brakishwater and fully marine taxa), red - freshwater (including freshwater and brakishwater taxa), yellow - terrestrial, blue - environment not recorded.

Click here for additional data file.

Supplemental Information 3 List of BINs possessing more than one ID variant with notes on the identification

BINs that have received different identifications with details of the ID and comments concerning the final identification used in the study.

Click here for additional data file.

Supplemental Information 4 List of BINs for which the environment was not able to be assessed

BINs for which the available data did not allow to specify the environment.

Click here for additional data file.

Supplemental Information 5 List of BINs with the largest number of records (blue - freshwater, green - marine, yellow - terrestrial taxa)

BINs that are represented by the largest number of records. The information about environmental origin of associated species are provided.

Click here for additional data file.

Supplemental Information 6 Nominal species with the largest number of BINs identified

Environmental origin of species is also provided.

Click here for additional data file.

Supplemental Information 7 List of amphipod families with number of accepted and barcoded species as well as information of the barcoding coverage within family

Families within each category with the highest barcoding coverage indicated in bold.

Click here for additional data file.

Supplemental Information 8 PRISMA 2020 checklist

Click here for additional data file.

Additional Information and Declarations

Competing Interests

Author Contributions

Data Availability

The authors declare there are no competing interests.

Anna Maria Jażdżewska conceived and designed the experiments, analyzed the data, prepared figures and/or tables, authored or reviewed drafts of the paper, and approved the final draft.

Anne Helene S. Tandberg conceived and designed the experiments, analyzed the data, authored or reviewed drafts of the paper, and approved the final draft.

Tammy Horton analyzed the data, authored or reviewed drafts of the paper, and approved the final draft.

Saskia Brix conceived and designed the experiments, authored or reviewed drafts of the paper, and approved the final draft.

The following information was supplied regarding data availability:

The list of records analysed is available in the Supplemental File. All data used are publicly available in BOLD.

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
