# Peer review of "Global gap-analysis of amphipod barcode library"

_PeerJ, doi:10.7717/peerj.12352_

## Round 0.1 · original submission · Minor Revisions

All three reviewers have indicated that your paper is acceptable, but requires some very minor revision.

Reviewer 1 ·

Basic reporting

I enjoyed reading this paper, and I believe it makes a timely and useful contribution to biodiversity, species recognition and DNA barcoding. The paper conforms with all journal criteria in this category. However, I strongly feel that the paper could be much more concisely written. As a random example, Line 357:

Ratnasingham & Hebert (2013) when presenting the BIN system indicated that overall 12% of the sequences available in BOLD at that time were lacking a family name, 19% a genus name and 40% a species name.

This sentence can be written much more concisely as follows: Ratnasingham & Hebert (2013) indicated that 12% of the amphipod sequences in BOLD lacked a family name, 19% a genus name, and 40% a species name

I believe that a careful revision of the paper along the lines above would make it much easier to read, and substantially reduce it's length

Experimental design

The paper meets the journal criteria outlined in this section

Validity of the findings

The data presented in the paper are straight forward, and the conclusions are supported.

Additional comments

Your paper is quite timely, and accurately highlights gaps in the amphipod barcode library. I suspect that many other taxa likely share the same geographic patterns of deficiency. Your paper would strongly benefit from revisions aimed a making the text much more concise.

·

Basic reporting

In this work, the authors compile a very detailed gap-analysis for an important invertebrate group. This type of work is very important in an era where DNA-based tools are being used for monitoring programs.

The paper is solid and good. The language is clear. The references are appropriate. The authors made a good job with clear goals, analysis and conclusions. I do not have major suggestions. The paper was very easy to read and review.

I think three minor issues may be clarified.

Line 216-219-> BAGS allows the user to choose. The user can remove records without data on country of origin or latitude or the user can download everything and do not apply this filter. Correct the phrase and explanation.

Line 232-> These 44 records were not removed? Supp S4 and lines 198-199 indicate that these records were removed.

Fig.1-> In the last box, “studies” in my opinion is confusing. Should be “records” no?

I also found some minor typos in lines 372,378, 407,499,501,539


(Without self-promoting) Two papers were recently published and may be interesting to add, if the authors wish so.


Line 121 and 407 -> Vieira et al 2021 could be a good citation if the authors wish
Vieira, PE, Lavrador, AS, Parente, MI, et al. Gaps in DNA sequence libraries for Macaronesian marine macroinvertebrates imply decades till completion and robust monitoring. Divers Distrib. 2021; 00: 1– 13. https://doi.org/10.1111/ddi.13305

Line 553-558-> Some initiatives for BOLD data curation are already happening
Radulovici AE, Vieira PE, Duarte S, Teixeira MAL, Borges LMS, Deagle B, Majaneva S, Redmond N, Schultz JA, Costa FO (2021). Revision and annotation of DNA barcode records for marine invertebrates: report of the 8th iBOL conference hackathon. bioRxiv 2021.03.07.434272. https://doi.org/10.1101/2021.03.07.434272

Experimental design

no comment

Validity of the findings

no comment

Additional comments

Congrats on the good work. It is rare to review "a final product" in the first stage of the reviewing process.

Reviewer 3 ·

Basic reporting

This manuscript by Jazdzewska et al. provides a descriptive analysis of barcode records for Amphipoda, Their results show that the barcoding coverage of amphipods is only 10%. Marine taxa had lower representation than freshwater taxa despite being more abundant. They also showed that some regions are less well studied than others (Africa and South America vs Europe and North America) and that several species rich families are not well represented.
This manuscript is straightforward and clearly written. The authors have carefully examined records of the bold database for analyses.
Sufficient background is provided but some relevant barcoding references were missing (ex. Tempestini et al. 2018, Duarte & Costa 2020).
The tables and figures are of professional quality.

Experimental design

This study is a descriptive one. The authors use a public database to provide a portrait of barcode gaps in Amphipoda. The authors claim that their objective was to provide a detailed understanding of strength and limitations of barcodes. I found that the manuscript had more to do with basic reporting of where most of the barcodes of Amphipoda came from and families that needed more coverage. I would have liked to see information on intraspecific divergence vs interspecific divergence in species from different habitats (marine, freshwater, terrestrial). Also, it would have been interesting to report amphipod records for different ocean basins.

Minor details
L250 something missing in the sentence
The authors often use the term 'environmental information' (for example l258). Collection locations would be more appropriate.

Validity of the findings

The underlying data are robust and sound. Statistical treatment does not apply here.
The discussion was focused mostly on amphipods. I suspect that similar results would be found in other orders of marine invertebrates. The conclusions are well stated and linked to the original question.

---

## Round 0.2 · accepted · Accept

I am content that your revisions have addressed reviewers' concerns, and that the paper is improved over the initial submission.